# Extensive Green Roofs (EGRs) and the Five Ws: A Quantitative Analysis on the Origin and Evolution, Aims, Approaches, and Botanical Views

Amii Bellini [1], Flavia Bartoli [1,2,*] and Giulia Caneva [1,3]

1 Department of Sciences, University Roma Tre, Viale Marconi 446, 00146 Rome, Italy; amii.bellini@uniroma3.it (A.B.); giulia.caneva@uniroma3.it (G.C.)
2 Institute of Heritage Science (CNR-ISPC), National Research Council of Italy, Area della Ricerca di Roma 1, Montelibretti, Via Salaria Km 29,300, 00015 Rome, Italy
3 National Biodiversity Future Center (NBFC), Università di Palermo, Piazza Marina 61, 90133 Palermo, Italy
* Correspondence: flavia.bartoli@cnr.it

**Abstract:** Extensive Green Roofs (EGRs) are nature-based solutions that provide several environmental, health, social, and economic benefits. This review of about 1430 scientific papers, based on the five Ws, *When*, *Where*, *Why*, *Who*, and *Which*, aims to understand how interest in these important green infrastructures originated and developed, as well as the nature of such academic research. Special attention was paid to the way researchers approached plant selection. Furthermore, this review made a detailed quantitative evaluation of the growth in interest for such green infrastructures within the scientific literature, which began mainly in Europe around the middle of the last century before spreading to America and Asia, growing rapidly during recent decades. The main impulse behind the study of EGRs came from the fields of engineering and architecture, especially on the themes of thermal mitigation and runoff reduction. In decreasing order, we found the categories aimed at ecological and environmental issues, substrate, and pollution reduction. We also found little evidence of collaboration between different disciplines, with the result that botanical features generally receive little attention. Despite the ecological benefits of plants, not enough attention has been given to them in the literature, and their study and selection are often limited to *Sedum* species.

**Keywords:** roof greening; urban ecology; biodiversity; urban planning; sustainable development

## 1. Introduction

The growth of urbanization in developed and developing countries is one of the most important social and economic phenomena in the overall development of different aspects of human life [1]. However, urban ecosystems are where interactions between anthropogenic activities and the natural environment are at their most intense [2,3]. Over the years, the expansion of cities has led not only to a reduction in green areas but also to greater environmental damage from the production of heat, waste, water, and air pollution and has generally had a negative impact on biodiversity [4–8]. There is an urgent need for integrative planning of green cities if we are to meet the growing environmental, social, and economic challenges posed by the negative effects of urban development [9]. Cities of the future should strive to be greener by designing and managing green infrastructure that can improve urban resilience and livability [10,11]. Even though urban forests and green open spaces in cities (tree-lined streets, gardens, parks, wetlands) are essential, there is often limited availability of ground-level spaces suitable for nature-based solutions.

Green roofs, typically consisting of vegetation planted above a series of layers that protect the roof substrate and improve the system's performance, can, therefore, represent an ideal supplementary space [12–14]. In particular, extensive green roofs (EGRs) require a shallower growing media and little maintenance compared to intensive green roofs (IGRs), which involve a thicker growing media and more intensive gardening [6].

Vegetated building roofing dates back to at least the Neolithic Era (8000–4000 BC), as they provided protection for buildings in harsh and climatically extreme environments [15]. For example, in the Arctic and the semi-arid continental lands of Central Asia, the scarcity of trees gave rise to a vernacular architecture where roofs were insulated with soil and living grasses from natural meadows (Figure 1a,b), known as "sod-roofs" [16]. The use of vegetation on buildings has also been recorded in the milder climates of medieval and modern Europe, particularly the beautiful, intensive green roofs built by aristocratic families and religious organizations [13]. These architectural structures gained complexity over the years with the invention of reinforced concrete, which allowed the construction of multi-story buildings with wide flat roofs and, as a result, provided many more opportunities for creating ornamental gardens above ground level [16].

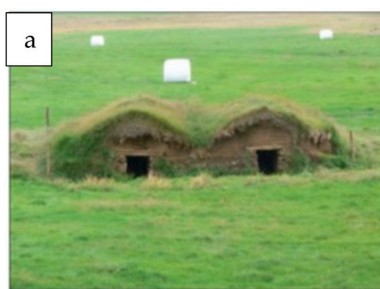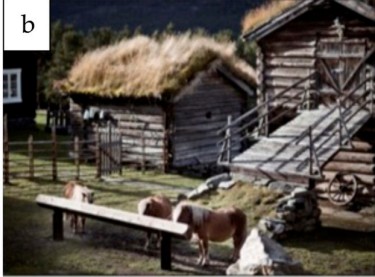

**Figure 1.** Old vegetated building roof (**a**) Traditional semi-subterranean sod house in Iceland (Photo credit Christian Bickel under Wikimedia Commons at "https://en.wikipedia.org/wiki/Sod_house". Accessed on 8 December 2023). (**b**) Traditional log farmhouses with sod roofs in Glittersjaa Mountain, Norway (Photo credit Tina Stafrén and "visitnorway.com". Accessed on 8 December 2023).

Studies carried out in Germany in the mid-twentieth century stimulated the birth of modern EGRs; thereafter, spontaneous vegetation growth was observed on flat roofs, demonstrating nature's ability to create green roofs on buildings [13]. Le Corbusier [17] was already talking about green roofs in 1927 and listed them among his 'Five Points of Modern Architecture', citing their functional benefits in protecting reinforced concrete from temperature changes, as well as their recreational value [18,19], while in 1936 the roof gardens created in South America by Roberto Burle Marx became famous [20].

Extensive green roofs have been the subject of several review papers and research papers, although these tended to focus on individual aspects [21–23]. Such studies have shown how EGRs help mitigate the urban heat island (UHI) effect, reduce temperature through evapotranspiration [12,14,24], and stormwater run-off by retaining water [12,14,25–27]. They absorb air pollution [14,28] and act as a sound barrier due to the thickness of the substrate and vegetation layers [29,30]. Green roofs also provide ecological benefits by supporting urban flora and fauna biodiversity and functioning as ecological corridors [6,31–34]. Finally, some revision work highlights how green roofs can affect different types of ecosystem services [35,36].

Although the historical and geographical background seems quite clear, interest in this topic has lacked consistency and homogeneity both in chronological and geographical terms. The same can be said for the themes chosen for study; although the spectrum of topics is very broad, interest in addressing them is not homogeneous. There is still a need for a more detailed analysis of 'where and when' interest in modern green roofs spread in the context of scientific research and 'why', i.e., with what goal such studies were made. Additionally, green roofs may be of interest to a wide range of disciplines, particularly engineering, architecture, agronomy, and botany. Another interesting question in need of clarification is 'who', i.e., the professional background of the study authors.

The botanical aspect of green roofs and their design, on the other hand, tends to be neglected, with little attention paid to plant selection. Although plant species play an important aesthetic role, which is a crucial architectural priority and important for psychological well-being [8,37], the plants selected are also fundamental to the functionality

of green roofs. For this reason, care should be taken to select the most appropriate plant species [6,38], and attention must be paid to the influence that soil depth, local climates, water availability, and planting density have on roof-based plants [19,39–41]

Based on a well-documented quantitative foundation that is lacking in previous articles and books published on the topic of green roofs, this review work was created with the aim of providing a useful tool when designing future experimental research. We have approached the subject of EGRs according to the principle of the five Ws of *When*, *Where*, *Why*, *Who*, and *Which*. The review encourages the reader to consider how certain issues have gained importance over time and in different places (*When* and *Where*), as well as highlighting any omissions in the issues addressed by EGR studies (*Why*) and the related disciplinary approach (*Who*). Finally, we highlight any issues related to the botanical aspect of plant species selection (*Which*).

## 2. Materials and Methods

### 2.1. Bibliographical Research

We obtained the relevant documents by searching digital scientific databases such as Google Scholar, Scopus, and Web of Science. These digital sources contain a more comprehensive range of content, nearly 50 million articles published since 1823 [42–44]. Our searches were performed targeting documents in all languages based on titles and abstracts containing the keywords "green roof", "roof greening", "roof garden", "vegetated roof", "planted roof" and "EGR farming", respectively to the publication language, published up to May 2022. After that, analyzing the references in the founded documents, we added old, hard-to-track, and no digital documents. For the master's and Ph.D. thesis, we considered only the documents produced by these researchers that were subjected to the peer-review process by experts to be sure of the scientific strength of the reported data.

### 2.2. Database Compilation and Method of Analysis

We grouped all documents in a digital database, which was structured according to the five Ws: *When*, *Where*, *Who*, *Why*, and *Which*:

(1) *When* (EGRs were studied)—In order to highlight how the scientific study of EGRs was distributed over time, we recorded the year of publication of each paper.

(2) *Where* (EGRs were studied)—Since we wanted to understand the geographical spread of interest in the study of green roofs and not the distribution of green roofs per se, we recorded the country of origin of the institution that conducted the research.

(3) *Why* (EGRs were studied), i.e., *What* was the intended benefit—We recorded why the author conducted the study, or rather what the authors' stated objectives were (e.g., to study aspects related to thermal mitigation, reduction of stormwater runoff, pollution reduction, an increase in biodiversity). In order to highlight predominant issues and those that were neglected, we collected relevant information from the Sections 3 and 6.

(4) *Who* (professional interest for EGRs)—In this case, we were interested in the professions of research team members (e.g., architects, botanists, engineers), so we relied on the information provided by the authors. We took note of all coauthors' affiliations to departments or research institutes.

(5) *Which* (selection criteria for plants in EGRs)—For the botanical aspect, we tracked information regarding plant species within the articles in order to highlight the attention authors paid to this aspect. In each case, we took into account the stated criteria (or lack of it) used for plant species selection (e.g., ecological, guidelines, undeclared).

We initially carried out a simple statistical analysis of the frequency of each of the above parameters before proceeding to perform a similar analysis on combined groups of parameters.

In cases where a paper presented more than one category in response to a W question, the paper was recorded under each of those categories.

For the *When*, we calculated annual publication trends from 1945 to 2022. For Where, we calculated the percentage of indexed green roof papers for each geographical area. For

*Why* (and *What*), *Which,* and *Who*, we created a number of categories corresponding to the various answers that emerged from our analysis and evaluated the percentage of papers within each category.

Finally, we also performed a PCA (principal component analysis) [45] on the relationships between categories belonging to '*why*', '*who*', and '*which*' (based on frequency data) to see whether there were common study criteria between different global regions. Specifically, we performed this type of analysis by grouping the collected items geographically, with one group for each of the five continents (Africa, America, Asia, Europe, and Oceania). If more than 100 items were collected for a single country, we added a separate group for that country.

## 3. Results

Our search produced a total of over 3500 peer-reviewed articles. After reading all titles and abstracts, we excluded documents with little relation to green roofs, for example, where studies cited green roofs as background information, reviews on general concepts, conference proceedings, or technical reports. Our final bibliographical list ended up including 1430 sources (in Supplementary Materials).

### 3.1. When (They Were the Object of Scientific Analysis)

The first recorded peer-reviewed scientific study on green roofs (therefore the earliest indexed in our database) dates back to 1945 and is by Kreh [46], who wrote about the phytosociology of such habitats in Germany. He points out that ruderal species adapt remarkably well to the colonization of the specific and extreme habitats of gravel roofs, providing insights, inspiration, and an impetus to move from the accidental to the intentional creation of functional green roofs on buildings [16]. Kreh's paper [46] and a series of subsequent studies published between the 1960s and 1980s in Germany and Switzerland (e.g., [16,47]) testify to the development of modern EGRs in these geographical areas.

90% of papers were published after 2010. Specifically, there was at least one paper on green roofs every decade from the 1940s except in the 1950s. Our database shows that until the first decade of the 2000s, a maximum of 9 papers were published each year on green roofs. In recent decades (from around 2008), interest in this topic has continued to grow, with a minimum of 14 papers per year, although the 100 mark was not surpassed until 2017, when another sharp rise was recorded. The yearly rate of publication between 2017 and 2020 remained below 113 until reaching a high of 155 annual publications in 2021. A total of 77 papers were recorded for the first half of 2022 (Figure 2).

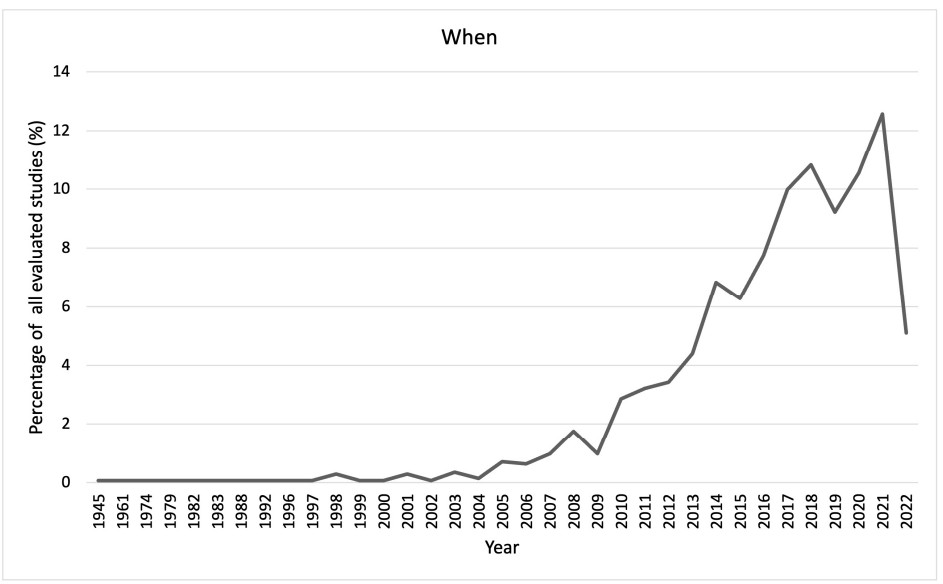

**Figure 2.** When graphs: Annual trend for publications over time.

### 3.2. Where (the Topic Has Been Studied)

Geographically, interest in this topic has spread across the globe (Figure 3a). The most productive research area was Europe, with 35% of all published articles, followed by Asia with 27% and North America with 20%. Publications from South America and Australia represented around 4% of the total, while those from Africa were below 1%. Despite total percentages at the continent level, the single country with the most publications is the United States of America, closely followed by China. As for Europe, the countries displaying the greatest interest in the study of green roofs were Italy at 9% and the United Kingdom at 5%, followed by Greece, Germany, and Spain with between 2.5% and 3.5% (Figure 3b). In general, countries with Mediterranean climates were seen to be very active in EGR studies and are among the most attentive to species selection.

**a**

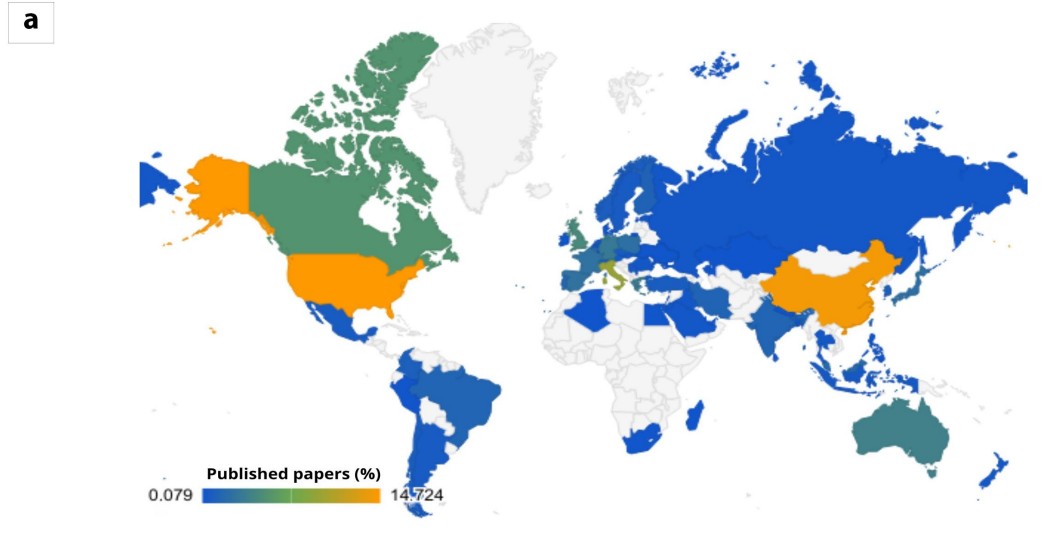

**b**

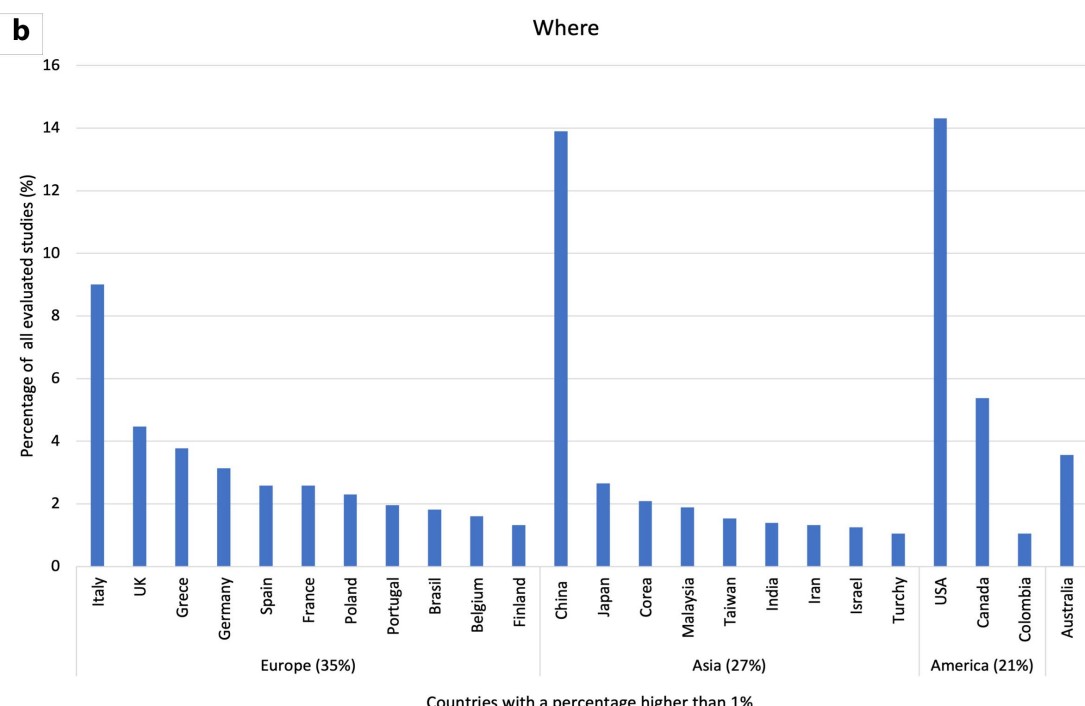

**Figure 3.** Where graphs: (**a**) distribution of indexed publications on green roofs and (**b**) percentage of all evaluated studies for each country with a value greater than 1%.

By combining the first two questions of When and Where, we can easily trace the origin and the spread of EGR studies. What stands out when cross-referencing data for time and place is the pioneering role played by Germany and Switzerland with regard to these issues. An analysis of the increase in interest in the study of green roofs over time and space reveals that such growth from the early 2000s and 2009 onward occurred in Central and Northern Europe. Beginning in the 1990s and continuing until 2015, we noted an increase in the number of European countries involved in such studies. Such involvement has since spread to more than 30 countries on all continents. In recent decades, the main contributors have been from North America and Europe, closely followed by Asia.

### 3.3. Why (Was It Studied) and What (Was the Benefit)

Regarding the topic of green roofs investigated by the researchers, eighteen topics related to green roofs were detected across all studies and can be summarized into the following seven macro-categories. Influences are to be considered both as determinant and as determinate (Figure 4b):

1. Thermal—includes all papers on UHI and thermal mitigation, energy saving, and environmental and climate improvement.
2. Hydrological—includes issues related to water, mainly stormwater runoff mitigation and green roof water management through irrigation.
3. Ecological—this category encompasses all naturalistic aspects, such as the role of increasing floristic diversity in creating ecological corridors and the benefits provided to wildlife species as well as issues related to the study of plant species (ecological and botanical studies) and to the ecosystem services provided by green roofs.
4. Substrate—The study of substrates, which may at first seem to fall within the scope of the ecological category, has been approached from an engineering as well as an ecological and engineering standpoint. For this reason, we have included it as a separate category covering agronomic factors related to substrate composition, the ability to support plant and animal species, water retention capacity, and weight and stratification features.
5. Pollution—this category includes topics related to the reduction of air or water pollution and studies that look at using green roofs as wastewater treatment and sound insulation systems.
6. Socioeconomic—includes all studies addressing human-related issues such as social, political, and psychological themes, like psychological well-being and effects on daily life. It also includes economic studies related to installation costs, maintenance, incentives, regulatory policies, and economic benefits.
7. Aesthetic/architectural—includes studies on the aesthetic appearance of green roofs, their use in urban redevelopment and planning, and aspects related to the design and safety of green infrastructures.

The broad nature of studies on this subject highlights the potential for interdisciplinary research on green roofs. Despite this, only 16% of papers involved two or more disciplines. The most common category was that of studies dedicated to engineering issues. In particular, thermal mitigation and runoff reduction are undoubtedly the most prominent themes; in fact, the thermal and hydraulic categories appeared with the highest frequency, accounting for 27% and 24% of studies, respectively (Figure 4b). The latter is followed by the combined category of ecological and environmental issues, appearing in 17% of studies. It is noteworthy that the substrate category alone, which is of interest to different research sectors, accounted for as much as 12% (Figure 4b). The pollution reduction category (air, water, and noise) was slightly less frequent and covered by about 10% of studies (Figure 4a). All other categories were present in about 5% of cases.

The studies' conclusions sections, after commenting on the success or failure of the trial, cited the main benefits as thermal mitigation and reduced stormwater runoff, with a frequency of 28% and 21%, respectively.

The improvement in green roof functionality cited by many papers (18%) was of a generic nature, and most of such papers were dedicated to the study of substrates. The only other benefit category with a high percentage was 'Ecology', which accounted for 15% of registrations, while all other categories scored between 4 and 9%. The benefit category with the lowest number of registrations was 'Aesthetics/Architecture'.

In general, the results of most studies found benefits from installing green roofs. Some negative factors emerged, mainly with regard to installation and maintenance costs. A small number of works refer to problems of water quality, observing that water runoff filtered through green cover layers, while seeing some reduction, is still rich in pollutants, some of which are the result of mineral nutrients and organics in the layers themselves.

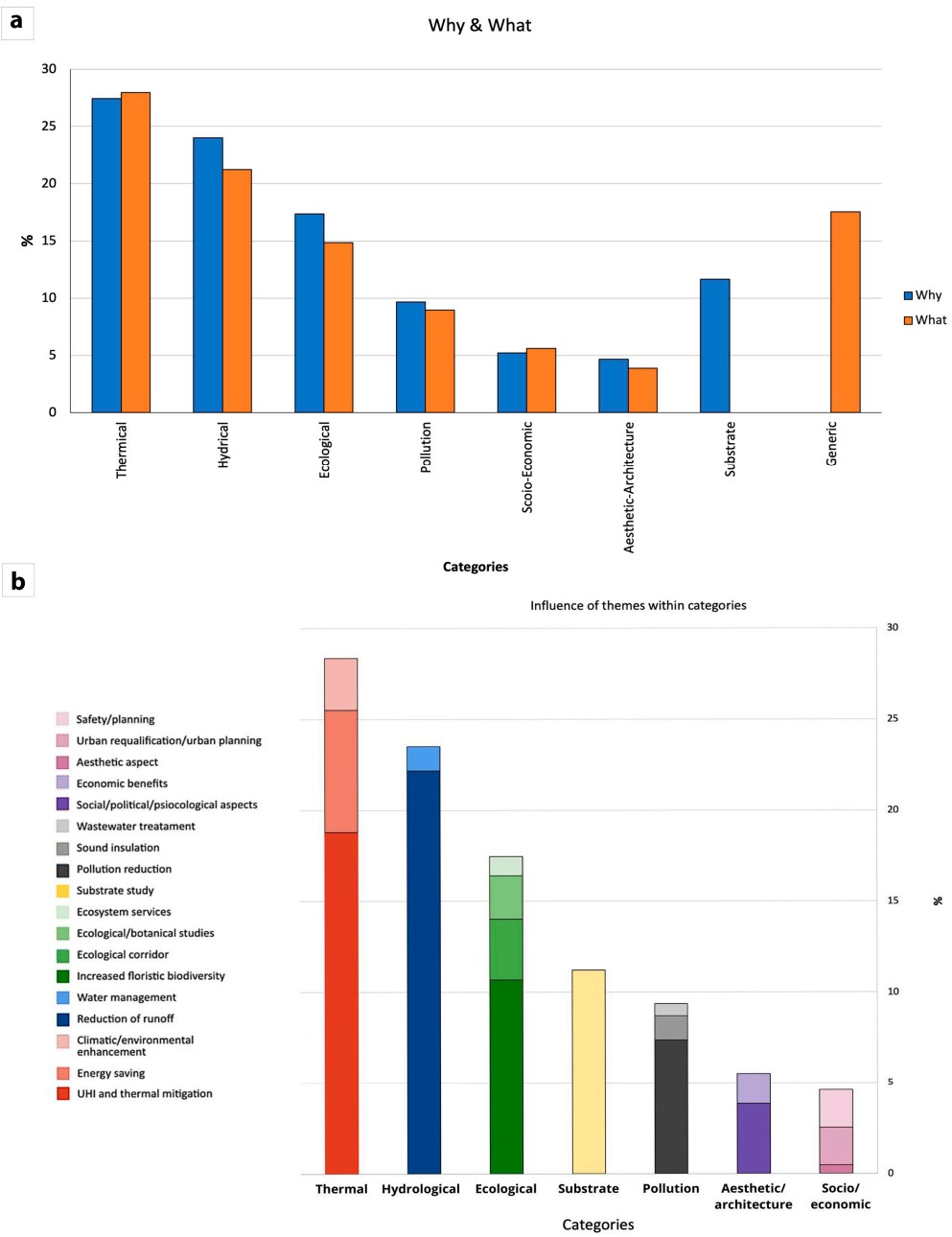

**Figure 4.** (**a**) Why and What graphs of uses for each category; (**b**) role of each theme within the single categories.

### 3.4. Who (Studied the EGR)

When looking at 'who' conducted the studies, we identified four categories of researchers. Category A contains research teams whose members are engineers and architects, while category B consists of foresters and agronomists, and botanists and naturalists in the broadest sense are grouped into category C. These are the categories that are most involved in the study of green roofs. We also added a more comprehensive category (D) for research teams made up of geographers, sociologists, and public research bodies. 95% of the studies were conducted by one of the core teams in categories A, B, and C (66.5%, 14.4%, and 19.2% of all papers, respectively) (Figure 5, with the D team accounting for only 5% and lacking any overlap from other categories. An interesting element observed in the three leading categories was a lack of interdisciplinarity. 96% of the studies were carried out by only one category of specialists (Figure 5), showing that multidisciplinary approaches to the study of green roofs are rare.

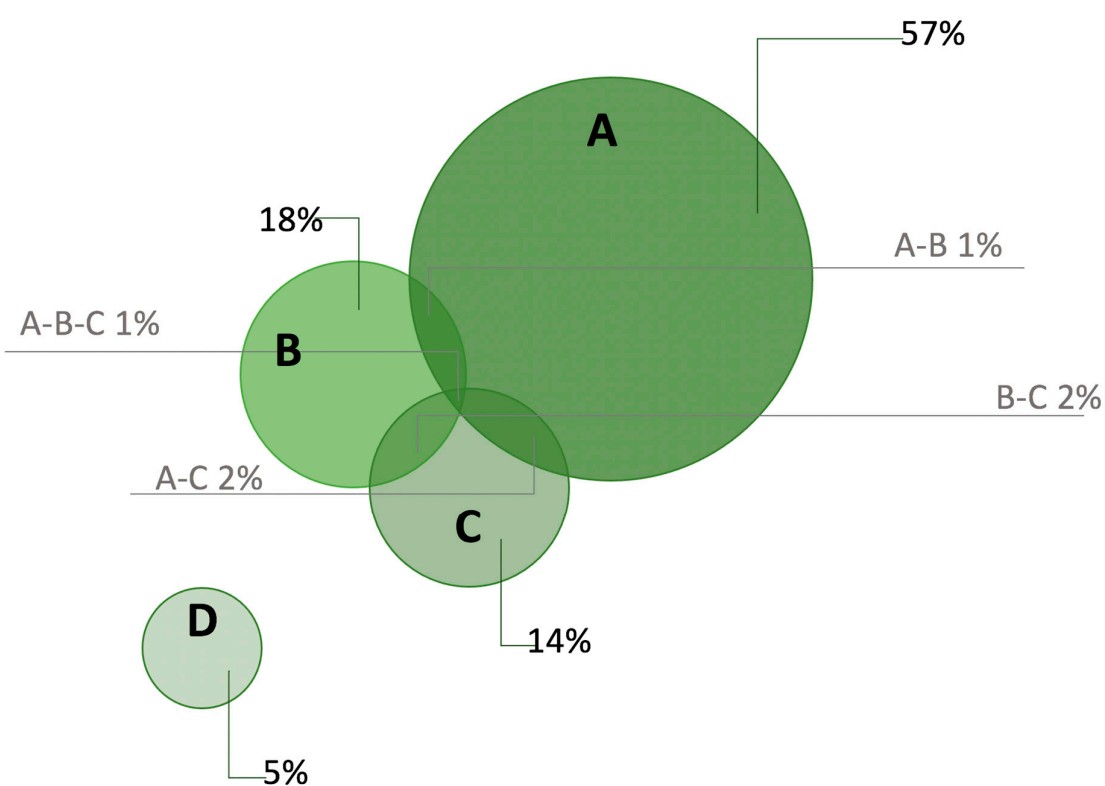

**Figure 5.** Who graphs of interdisciplinarity percentage ((A) = engineers or architects, (B) = foresters or agronomists, (C) = botanists or naturalists, (D) = geographers, sociologists, or state research organizations).

### 3.5. Which (Selection Criteria Was Used for Plants)

A total of 15 different plant selection criteria emerged from authors' declarations (Unspecified, Commonly Used/Following Guidelines, Ecology, Biogeography, Adaptability to Pioneer Situations, Aesthetic/Ethnobotanical Value, Morphology, Low Maintenance, Bioclimate, Habitat Pattern, Easy Availability, Phytosociology, Supporting Urban Biodiversity, Physiology, Phenology). As with the themes that emerged in the 'why' section, we summarized these criteria into the following six macro-categories defining plant selection criteria (Figure 6a):

1. Ecology—encompasses all strictly ecological themes such as ecology, bioclimate, habitat patterns, adaptability to pioneer situations, and phytosociological similarity.
2. Biogeography—referring mainly to species native to that area.
3. Technical-Practical—includes themes such as easy availability, low maintenance, aesthetic/social value, and the promotion of urban biodiversity.

4.  Following Guidelines—documents in which authors selected species according to various guidelines or to common usage.
5.  Morpho-physiology—contains the topics of morphology, phenology, and physiology.
6.  Unspecified—includes papers with no reference to plants or selection criteria.

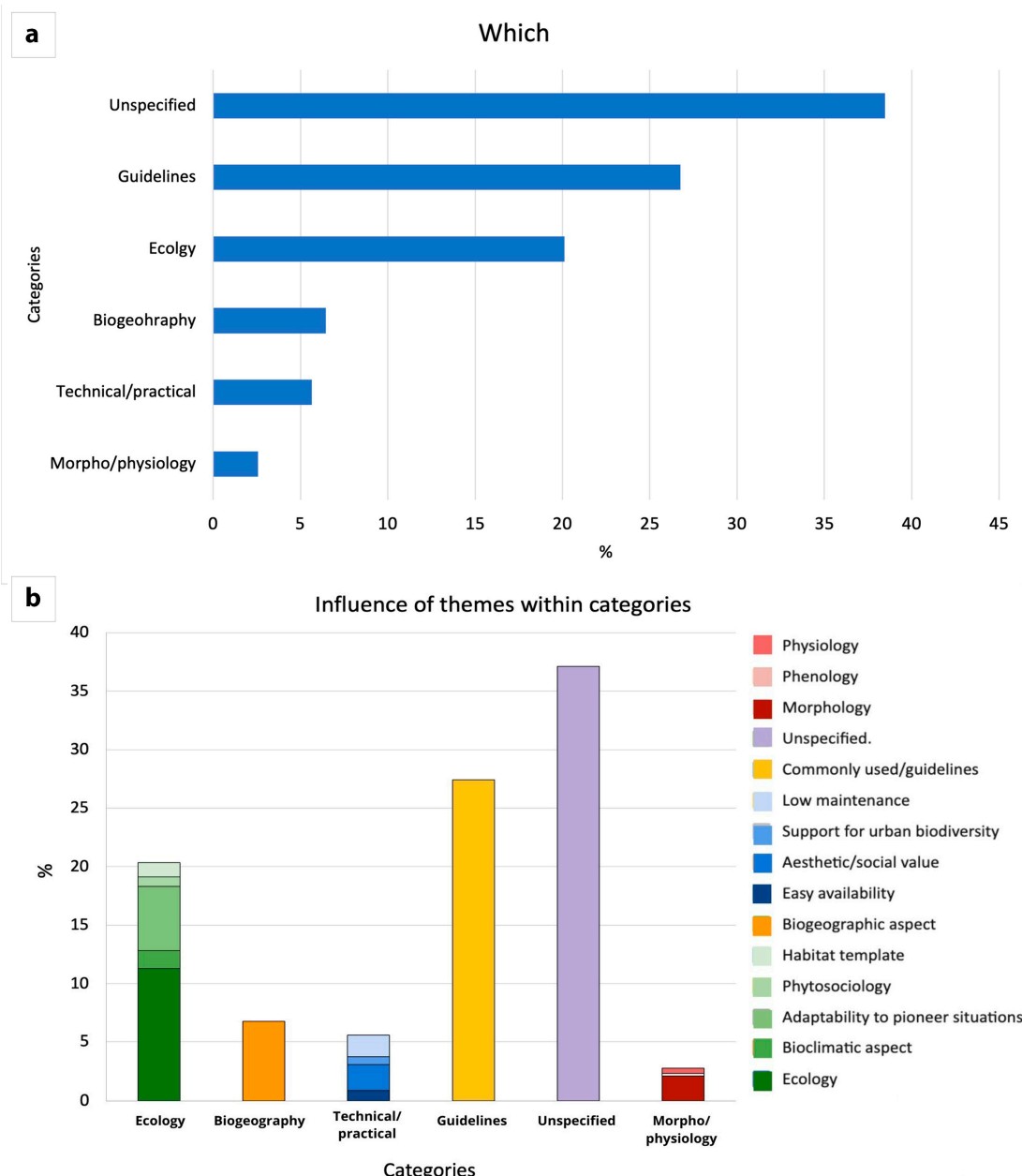

**Figure 6.** Which selection criteria were used for plants. (**a**) Percentage of different categories; (**b**) Frequency of use for each category.

Surprisingly, 'Unspecified' was the largest single category, covering 37.9% of all articles in our study. Four of the six categories accounted for 35.1% of studies. Of these four, the most common was 'Ecology', due in large part to the frequency of research on species' biological forms, followed by 'Biogeography'. The 'Following Guidelines' category represented 27.0% of papers (Figure 6b). Many studies refer to plants as "the most frequently used vegetation in green roofs" or give the plant genus *Sedum* without specifying the species. In other cases, they referred simply to "*Sedum*" or even "grass-type plants". In general, *Sedum* species were the most common regardless of the reason for selection. Although *Sedum* sp. was recorded in almost all papers that followed the 'Following Guide-

lines' category, other reasons were cited, including resistance to water, heat, or edaphic stress conditions, the need for low maintenance, a pronounced ability to spread and cover the soil and a harmonious aesthetic appearance.

### 3.6. The Relationship between Ws

In addition to the main groups related to the five continents, we added three more groups for countries with more than one hundred published studies (China, Italy, and the USA).

Three groups emerged from the Principal Component Analysis (Figure 7). Group I is composed of Italy, China, America, and Asia; group II of the USA, Europe, and Oceania; and group III represents Africa.

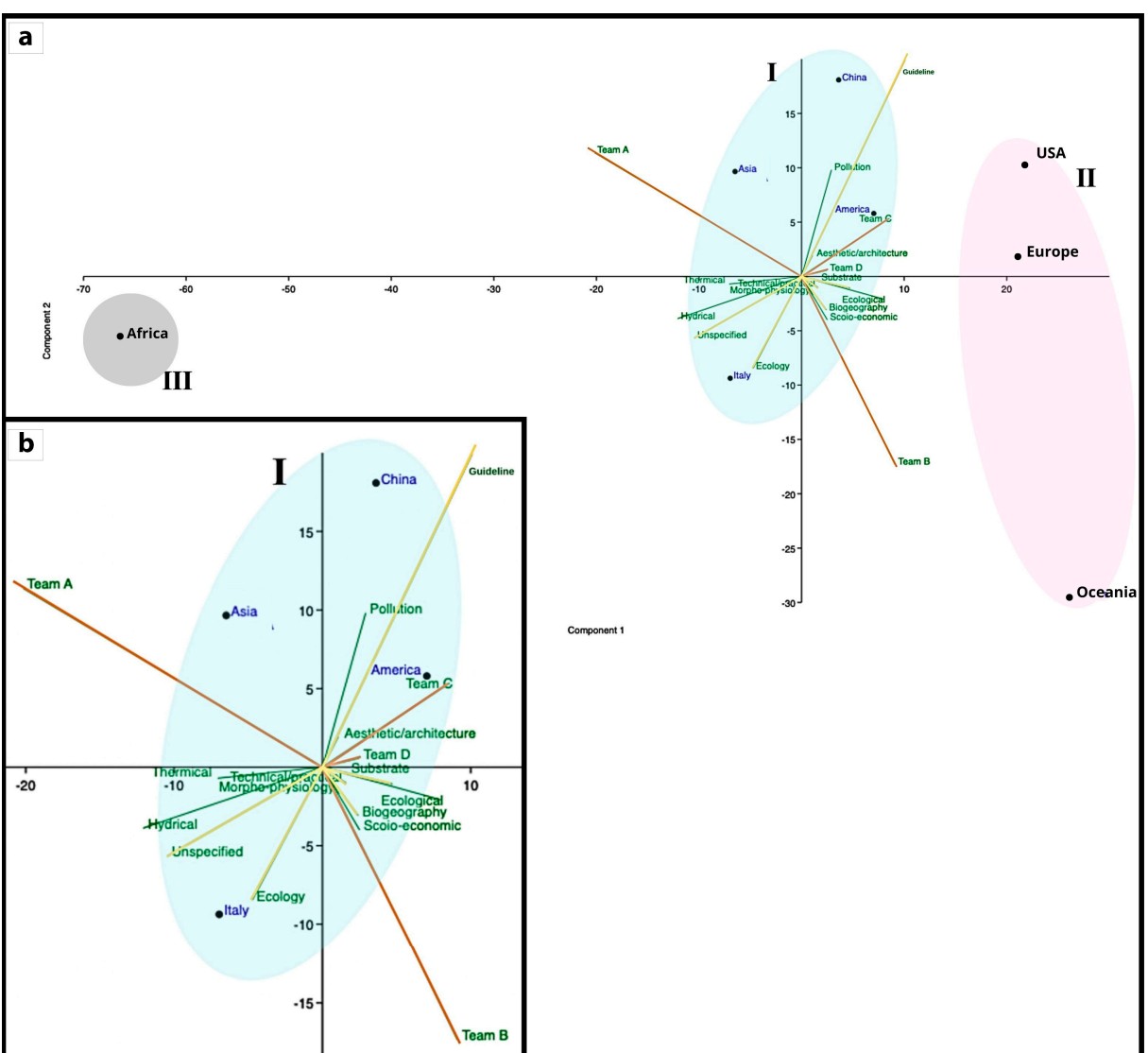

**Figure 7.** (**a**) PCA analysis performed with the frequency of use of categories belonging to Why, Who, and Which. (Teams: A = engineers or architects, B = foresters or agronomists, C = botanists or naturalists, D = geographers, sociologists, or public research bodies). (**b**) Detail of Group I.

The characteristics of Group I were generally intermediate, although Italy tends more towards using ecological features as a criterion for species selection, while America and China favor the use of guidelines. Asia's most distinguishing feature, on the other hand, is the predominance of professional figures belonging to Team A.

Group II is similar to Group I; again, we see how the US mainly used guidelines and studies related to pollution; in Europe, there was a significant presence of professional figures from Team C, while in Oceania, Team B professionals were very active.

Group III stands apart with no unique characteristics other than an interest in engineering issues such as temperature and hydrological features.

## 4. Discussion

### 4.1. Origin and Evolution

Although the principles of EGRs can be found in early twentieth-century modern architecture, we can say with certainty that their introduction and related research began some decades later. This probably happened, like with many other innovations, as a result of the complex socioeconomic and political conditions created by the two world wars. There is now a truly global interest in research into this subject driven by curiosity, fashion, and necessity.

With regard to the geographical origin of EGRs, our vast database confirmed the role of Germany and Switzerland in early implementation and scientific research, which then expanded worldwide [48,49]. There is a margin of error in the precise quantification of scientific papers in the field since the indexing of papers began later and was implemented differently in different countries. Recent and rapidly spreading interest in places like Southern Europe, China, and the United States may be associated with an increased awareness of environmental issues and economic benefits [50]. Indeed, over the past three decades, environmental degradation has become a source of collective concern, leading many national and international organizations to launch specific initiatives based on sustainability in order to counter the progress of climate change and environmental degradation [51,52].

However, despite the need to address these issues collectively, countermeasures are not taken in the same way and at the same time by all nations, and cultural, political, and economic factors can influence how nations approach these strategies. This divergence is also clear in the present study.

The challenges related to political decisions and the market are evident, even when considering countries that emerge as the most active in this context, such as the United States, China, and Europe, where green roofs have become a central theme. Despite the establishment of organizations like GRHC (Green Roofs for Healthy Cities) or EFB (European Federation Green Roof and Walls) with the aim of promoting and encouraging the adoption of green roofs and walls in their respective countries, it is observed that regulatory frameworks and political incentives remain insufficient. Additionally, there is a lack of in-depth analysis and research on the management of the green roof market. For instance, although China is recognized as a global leader in constructing new areas every year, it still lacks adequate regulations on green roofs to stimulate their development [53].

### 4.2. Aims and Approaches

From the earliest published papers, we found that despite being a multidisciplinary field, the study of green roofs is too often treated in a monodisciplinary manner [49,54].

The only negative finding common to all the papers we analyzed was related to costs. This confirms the conclusions of Chen et al. [55] and Dong et al. [50], who identified maintenance, design, and construction costs, poor provisions for EGR adoption, and lack of subsidies as the main disincentives to the implementation of EGRs.

A general evaluation of the various approaches suggests little consideration was given to ecological concerns. Usually, when engineer/architect teams specify the selection method for plants, their decisions are made mostly on a technical or practical basis by considering morphology, canopy capacity, aesthetics, easy availability, and low maintenance. The terminology used is another element that highlights the need for more emphasis on plant selection. Authors often refer to vegetation with generic and minimalist terms such as 'grass', 'grass-like', and '*Sedum*-like'. This denotes how, in some cases, vegetation is seen

superficially as purely aesthetic or as an accessory. Ecological and environmental factors have recently been receiving more interest due to the role of biodiversity in environmental management as a hot topic, even in urban contexts [49,56–59]. This can also be seen in national and international policy trends. One such policy is the recently adopted Biodiversity Strategy 2030 [60], which aims to "bring nature back into our lives", in line with the goals of the Green Deal [61].

*4.3. Botanical View*

In most countries, from the United Kingdom and Germany to America and China, *Sedum* species (Crassulaceae) are the plants most used. Their selection is justified by the belief that they are the most suitable species for this environment, and they are often recommended in guidelines. It is undeniable that species selection is a challenge, and selecting drought-resistant species such as *Sedum* sp. on extensive green roofs is easy for any professional figure to justify [50,56,62].

There is increasing awareness of the need to employ a wider range of *Sedum* species by considering biogeographical and bioclimatic factors, using native species [63,64], or considering physiological and ecological factors like prioritizing species able to tolerate water stress [65,66], possess specific photosynthetic qualities (e.g., CAM, C4) [66,67] or can adapt to pioneer conditions [68,69]. However, the progress of research is slow. For example, only about a hundred species appear in papers published in North America despite its extreme environmental complexity and wide range of climatic conditions [48]. From an ecological point of view, we should be aware that *Sedum* species are not native to some parts of the world, and research should also focus on considering the suitability of other plant species for green roofs [54]. *Sedum* could be replaced with other species suited to the green roof construction environment by using ecological species selection methods [6]. In this way, by considering the variety of functions performed by different plant forms, green roofs can be designed so as to promote conditions beneficial to plants and maximize benefits [60,70].

Species diversity in green roofs has often been seen as simply an aesthetic issue [71]. Time and again, plant diversity has been shown to play an important role in the functionality of these infrastructures. For example, it can improve substrate cooling [72], prevent invasive weeds [73], and conserve water [54,74]. Plants are also crucial to social, psychological, and sometimes even to ethnobotanical considerations [75–78]. In some cases, for example, green roofs are used to grow food and treat water [79–82].

## 5. Further Needs and Recommendations

Based on our findings, we have developed some recommendations for improving the effectiveness of research, for the prevention of setbacks and to strengthen connections between field application, regulations, and professional practice.

1.  Interdisciplinarity: This topic would benefit from a multidisciplinary approach, but our study reveals that research teams are overwhelmingly monodisciplinary. Therefore, it is essential to encourage multidisciplinary so that research teams are able to take all necessary factors into consideration. Moreover, research has always focused predominantly on engineering, architecture, and construction, often implementing software-based modeling and simulations rather than actual experiments, which are fundamental to an ecological and botanical approach.
2.  Ecological sensibility: Ecological factors (i.e., the study of the environment and the selection of plant species) should be considered paramount when dealing with vegetated roofs but are often neglected or underestimated. In light of the benefits that green roofs can provide and the importance of their ecological features for the functioning of the roof itself as well as that of the urban ecosystem, greater ecological sensitivity is needed. These considerations can significantly increase the efficiency of green roofs and their economic and environmental sustainability.

3.  Guidelines: Countries that have not developed adequate research on which species to use generally refer to the FLL guidelines [82]. However, such guidelines are aimed only at specific types of environments. Work is being conducted to develop local guidelines, but no document comparable to the FLL guidelines is currently available [48,83,84]. A very high percentage of studies make use of guidelines designed for a specific geographical area with different environmental conditions from the one they are working in, especially with regard to plants. Such guidelines, rather than providing a list of species, should provide the tools to facilitate their selection based on the physiological, morphological, and ecological characteristics that allow them to survive in a particular environment. Specifically, selection should focus on analyzing regional and local climates, including rainfall patterns and building location, as well as considering solar irradiance, currents, and substrate.

4.  Financial support: Finally, we recommend following the policies enacted in Switzerland and Germany, where incentives, regulations, and guidelines have been introduced to facilitate the implementation of green roofs. Such policies facilitate the application and maintenance over time of these essential nature-based solutions [85–88].

## 6. Conclusions

This first quantitative review, based on a detailed analysis of the international literature, shows that the EGRs topic, which had a great impulse in the last decades starting from the first European experiences, was mainly approached with an engineering and architectonic aim due to their functional benefits, with a very low level of interdisciplinary collaboration (less than 5%). The main benefits recorded from the engineering studies were thermal mitigation and reduced stormwater runoff. The topic of plant selection was, indeed, insufficiently analyzed, and the reason for plant selection is very scarcely documented. A botanical approach, which can widen species selection, now dominated by *Sedum* species, can bring many benefits and make the green roof more functional and environmentally sustainable. We provide recommendations for considering such topics with a multidisciplinary focus, considering that engineering and architectural issues should be analyzed with ecological aspects to improve the true sustainability of the project.

**Supplementary Materials:** The following supporting information can be downloaded at: https://www.mdpi.com/article/10.3390/su16031033/s1, Table S1: List of all the 1430 analyzed.

**Author Contributions:** G.C. and A.B. have particularly contributed to the study conception and design; Methodology: G.C., A.B. and F.B.; Formal analysis and investigation: A.B.; Writing—original draft preparation: G.C. and A.B.; Writing—review and editing: All Authors; Funding acquisition: G.C.; Supervision: All Authors. All authors have read and agreed to the published version of the manuscript.

**Funding:** Grant of Excellence Departments, MIUR-Italy Italian Ministry of University and Research, National Operational Program "Research and Innovation" 2014–2020 (PON), CUP ECCELLENZA_2023-27_BMCAE. National Biodiversity Future Center (NBFC), Università di Palermo, Piazza Marina 61, 90133 Palermo, Italy.

**Institutional Review Board Statement:** Not applicable.

**Informed Consent Statement:** Not applicable.

**Data Availability Statement:** Not applicable.

**Acknowledgments:** We would like to thank Luca D'Amato for his kind help with the statistical elaborations and Justin Bradshaw for his help with the English translation.

**Conflicts of Interest:** The authors declare no conflicts of interest.

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
