# Peer review of "Extensive Green Roofs (EGRs) and the Five Ws: A Quantitative Analysis on the Origin and Evolution, Aims, Approaches, and Botanical Views"

_sustainability, doi:10.3390/su16031033_

Round 1
Reviewer 1 Report
Comments and Suggestions for Authors
The bibliographic work is carried out correctly, I found the analysis well explained and very usable, in particular I appreciated the creation of the database according to the 5Ws and the subsequent sub-categorisations, I also really liked the historical framework, very well done and useful for researchers who are not used to framing their research historically and anthropologically. However there is a small fixes that need to be made to have a truly great review.
The first and most important concerns Figure 2, in particular you can easily realize that from 1945 to today the methods of publication (from paper to digital), the number of people on the planet (and therefore the number of researchers) and the number of papers published in different fields (Architecture, Botany, Engineering, Ecology, etc...) therefore those values you give in the graph should be represented as a percentage compared to something else i.e. the number of papers published per year in the sectors you have analysed. It is important that the data in figure 2 is rescaled and represented as a percentage, explaining the choice and the reason for the percentage with respect to what is chosen.
I also suggest trying to make images and graphs more readable, it would be a good idea for the font size in images and graphics to be of a comparable size to the font size in the text. In particular, in Figure 1 the stratification is difficult to read, in Figure 3 the states are difficult to read and I suggest putting the writing in black and not grey, I also suggest representing the scientific production of the entire EU as yet another bar.
Same thing in figures 4, 6 and 7, the writing is difficult to read.
I also suggest better developing the state of the art, especially by better indicating and summarizing the political directions and realities of the sector in at least the USA, Italy (EU) and China.
Allow me to suggest some sources that I believe could enrich the work if consulted:
https://doi.org/10.1016/j.rser.2014.07.147
https://efb-greenroof.eu/
https://greenroofs.org/
In particular, it is important to remember that for this topic the scientific interest is closely linked to political decisions and market sentiments.
Author Response
The bibliographic work is carried out correctly, I found the analysis well explained and very usable, in particular I appreciated the creation of the database according to the 5Ws and the subsequent sub-categorisations, I also really liked the historical framework, very well done and useful for researchers who are not used to framing their research historically and anthropologically. However there is a small fixes that need to be made to have a truly great review.
A: Thank you for your appreciation
The first and most important concerns Figure 2, in particular you can easily realize that from 1945 to today the methods of publication (from paper to digital), the number of people on the planet (and therefore the number of researchers) and the number of papers published in different fields (Architecture, Botany, Engineering, Ecology, etc...) therefore those values you give in the graph should be represented as a percentage compared to something else i.e. the number of papers published per year in the sectors you have analysed. It is important that the data in figure 2 is rescaled and represented as a percentage, explaining the choice and the reason for the percentage with respect to what is chosen.
A: Thank you very much for your valuable advice. We have taken into consideration your suggestion, and following your recommendation we converted the data into a percentage calculated on the total number of analyzed articles.
I also suggest trying to make images and graphs more readable, it would be a good idea for the font size in images and graphics to be of a comparable size to the font size in the text. In particular, in Figure 1 the stratification is difficult to read, in Figure 3 the states are difficult to read and I suggest putting the writing in black and not grey, I also suggest representing the scientific production of the entire EU as yet another bar.
Same thing in figures 4, 6 and 7, the writing is difficult to read.
A: thank you for your valuable suggestion. We worked to enhance the quality and readability of all the images, following your guidance.
I also suggest better developing the state of the art, especially by better indicating and summarizing the political directions and realities of the sector in at least the USA, Italy (EU) and China. Allow me to suggest some sources that I believe could enrich the work if consulted:
https://doi.org/10.1016/j.rser.2014.07.147
https://efb-greenroof.eu/
https://greenroofs.org/
In particular, it is important to remember that for this topic the scientific interest is closely linked to political decisions and market sentiments.
A: thank you for your valuable suggestion. Following your guidance, we added a paragraph on this topic, respectively with adding the reference suggested and for the website we add the URL directly in the text as link.
Reviewer 2 Report
Comments and Suggestions for Authors
The work "Extensive Green Roofs (Eggs) and the Five Ws ..." fully corresponds to the subject of the journal and is addressed to an urgent problem. The work is written in good language, conveniently structured, supplied with a reasonable number of good illustrations that successfully complement the text of the article. The references are complete enough for an ordinary article of this kind. The abstract satisfactorily outlines the research topic, the conclusions are complete, justified and contain interesting and scientifically new conclusions. For example, the conclusion about the observed anomalous lack of interdisciplinarity is interesting. The reviewer did not notice any serious comments or errors.
The article is valuable and can be published in the journal in its current form, or slightly corrected (the decision is at the discretion of the authors) in accordance with some considerations about the methodology used, given below.
The reviewed work is interesting primarily by the methodology used - the subject of research is not the scientific problem itself, but the reflection of this problem in the mirror of numerous published scientific papers. Such an approach is rare in the practice of modern research, but it is undoubtedly fruitful and interesting. The division of the problem into 5W is debatable, but in any case it is acceptable, especially in connection with the uniqueness of the study. Some problematic areas of the study should be noted for further improvement.
1. The study was conducted on the basis of 1,280 scientific papers, while only 87 positions are indicated in the references. This discrepancy significantly limits the verification of the study. It is highly desirable to mention all 1280 works in some form - maybe not in the reference, but in the attached file.
2. Using only English to search for publications in index databases leads to obvious bias. For example, Brazil has a solid and long-standing experience in implementing the EGR approach in megacities, but the contribution of this country to this study is negligible - probably many aspects of the problem were described in publications in Portuguese. The use of different languages is necessary in the global analysis of publications in the future - although the problems are obvious.
Author Response
The work "Extensive Green Roofs (Eggs) and the Five Ws ..." fully corresponds to the subject of the journal and is addressed to an urgent problem. The work is written in good language, conveniently structured, supplied with a reasonable number of good illustrations that successfully complement the text of the article. The references are complete enough for an ordinary article of this kind. The abstract satisfactorily outlines the research topic, the conclusions are complete, justified and contain interesting and scientifically new conclusions. For example, the conclusion about the observed anomalous lack of interdisciplinarity is interesting. The reviewer did not notice any serious comments or errors.The article is valuable and can be published in the journal in its current form, or slightly corrected (the decision is at the discretion of the authors) in accordance with some considerations about the methodology used, given below.
A: Thank you, for the appreciation
The reviewed work is interesting primarily by the methodology used - the subject of research is not the scientific problem itself, but the reflection of this problem in the mirror of numerous published scientific papers. Such an approach is rare in the practice of modern research, but it is undoubtedly fruitful and interesting. The division of the problem into 5W is debatable, but in any case it is acceptable, especially in connection with the uniqueness of the study. Some problematic areas of the study should be noted for further improvement.
The study was conducted on the basis of 1,280 scientific papers, while only 87 positions are indicated in the references. This discrepancy significantly limits the verification of the study. It is highly desirable to mention all 1280 works in some form - maybe not in the reference, but in the attached file.
A: Thank you for your suggestion. We have attached the comprehensive list of analyzed works as supplementary material as Appendix 1.
- Using only English to search for publications in index databases leads to obvious bias. For example, Brazil has a solid and long-standing experience in implementing the EGR approach in megacities, but the contribution of this country to this study is negligible - probably many aspects of the problem were described in publications in Portuguese. The use of different languages is necessary in the global analysis of publications in the future - although the problems are obvious.
A: thank you for the suggestion, we used English keywords, but the search engine revealed the keywords respectively in the language of the publication, indeed we found all the paper also of other languages. In any case, we modified in the manuscript clearing the sentences.
Reviewer 3 Report
Comments and Suggestions for Authors
Bibliographical Research: The researcher obtained the relevant papers by searching the Scopus database only.
The criteria for selecting should be clear, and what about the period before creating the Scopus indexing?
And what about the research published in other non-Scopus journals or the master's or Ph.D. thesis too?
The research problem was not explained enough, and the objectives should be cleared too.
Avoid using [we]
Author Response
Bibliographical Research: The researcher obtained the relevant papers by searching the Scopus database only. The criteria for selecting should be clear, and what about the period before creating the Scopus indexing? And what about the research published in other non-Scopus journals or the master's or Ph.D. thesis too? The research problem was not explained enough, and the objectives should be cleared too.
A: Thank you for you suggestions, we cleared in the main text that we searched in Scholar, Scopus and Web of Science to have a overall view of the scientific literature published. Moreover, by the reading of the references of the found digital paper we discovered and considered the oldest document not digital. For the Master and PhD thesis we considered only the published data, because subjected to peer review process by experts, in this way we can be sure of the strongness of the data. In any way, we rearranged the section “2.1 Bibliographic research” to clarify these points.
Reviewer 4 Report
Comments and Suggestions for Authors
Interesting paper relevant to the contemporary trend of sustainability and green buildings.
Abstract well written, all necessary parts well described
Main issue, when preparing literature review / bibliometric study, the WoS database that beside CC contains various indexes should be explored too
It is not enough only to present the data from Scopus database and this should be corrected.
After adding WoS biblimetric research, that the topic will be well explored and the contribution will have scientific soundness.
I highlighted this major problem with this interesting paper, I don't find other issues but this one is the major.
I cannot accpet scientific paper focused only on Scopus database.
If the both databases (WoS and Scopus) were included, then I would immediately accept the paper without any comments.
The paper is good, well structured (by my opinion) but the scientifc soundness is lacking because only Scopus database was researched.
I belive that it is not necessary to write long sentences and specific comment comments for the aspect that I dont find problematic. Additionaly, the reviewer form is not structured in this way.
Comments on the Quality of English Language
English language is in order.
Author Response
Interesting paper relevant to the contemporary trend of sustainability and green buildings. Abstract well written, all necessary parts well described.
A: Thank you for you appreciation
Main issue, when preparing literature review / bibliometric study, the WoS database that beside CC contains various indexes should be explored too. It is not enough only to present the data from Scopus database and this should be corrected. After adding WoS biblimetric research, that the topic will be well explored and the contribution will have scientific soundness. I highlighted this major problem with this interesting paper, I don't find other issues but this one is the major.I cannot accepet scientific paper focused only on Scopus database. If the both databases (WoS and Scopus) were included, then I would immediately accept the paper without any comments. The paper is good, well structured (by my opinion) but the scientifc soundness is lacking because only Scopus database was researched. I belive that it is not necessary to write long sentences and specific comment comments for the aspect that I dont find problematic. Additionaly, the reviewer form is not structured in this way.
A: Thank you for your valuable suggestion. We have expanded our research to include Web of Science (WoS) as well. In general, a comparable number of works emerged, with the majority overlapping with those already analyzed from Scopus. The few differing articles have been scrutinized and incorporated into the study.
Round 2
Reviewer 1 Report
Comments and Suggestions for Authors
Concerning figure 2 my suggestions has been misintepreted, what I mean is that if in 1980 10 papers were published in the sector of interest and only 2 deal with green roofs the percentage should be 20%, if in 2015 10,000 papers were published in the sector of interest and only 100% deal with green roofs the percentage is 1%.
This bias in bibliographic and bibliometric analyzes is well known and must be taken into account. With the advent of new technologies, the publication of scientific articles has increased dramatically. What assures us that the increase you observe on green roofs related topic is not due to this more than to actual scientific interest?
Author Response
Dear reviewer, thank you for your suggestions. however, we feel that the interest of the different sectors in green roofs has been highlighted in the following graphs (4,5), where the percentage of interest from the different disciplines and professionals has been analysed.
Undoubtedly, the considerable increase in publications linked to the advent of new technologies has also led to a steady increase in literature related to green roofs. However, as highlighted in the discussion of all the results, what emerges even from the simple percentage of articles published per year, such as the current Figure 2, is that the increases in interest over time have been marked by historical moments and socio-political changes such as: enactment of laws and incentives, environmental awareness, and the need to solve problems. These innovations and incentives have led to tangible increases in scientific interest in the subject.
Furthermore, as better explained in section 2.1 during the last review, although most of the analysed papers are derived from digital databases such as WoS and Scopus, our list of papers also contains grey bibliography, hard-to-find and non-digital documents that therefore make the suggested comparison more difficult.